# Identifiability in Noisy Label Learning: A Multinomial Mixture Modelling Approach

## Abstract

Learning from noisy labels (LNL) is crucial in deep learning, in which one of the approaches is to identify clean-label samples from poorly-annotated datasets. Such an identification is challenging because the conventional LNL problem, which assumes only one noisy label per instance, is non-identifiable, i.e., clean labels cannot be estimated theoretically without additional heuristics. This paper presents a novel data-driven approach that addresses this issue without requiring any heuristics about clean samples. We discover that the LNL problem becomes identifiable if there are at least $2C - 1$ i.i.d. noisy labels per instance, where $C$ is the number of classes. Our finding relies on the assumption of i.i.d. noisy labels and multinomial mixture modelling, making it easier to interpret than previous studies that require full-rank noisy-label transition matrices. To fulfil this condition without additional manual annotations, we propose a method that automatically generates additional i.i.d. noisy labels through nearest neighbours. These noisy labels are then used in the Expectation-Maximisation algorithm to infer clean labels. Our method demonstrably estimates clean labels accurately across various label noise benchmarks, including synthetic, web-controlled, and real-world datasets. Furthermore, the model trained with our method performs competitively with many state-of-the-art methods.

## 1 Introduction

The significant advances in machine learning over the past decade have led to the development of numerous applications that tackle increasingly-complex problems in many areas, such as computer vision (Krizhevsky et al., 2012; Dosovitskiy et al., 2021), natural language processing (NLP) (Bahdanau et al., 2015; Vaswani et al., 2017) and reinforcement learning (Silver et al., 2016; Jumper et al., 2021). These solutions often rely on high-capacity models trained on vast amounts of annotated data. The annotation of large datasets often relies on crowd-sourcing services, such as Amazon Mechanical Turk, or automated approaches based on NLP or search engines. This might, generally, produce poor-quality annotated labels, especially when data is ambiguous. Such a subpar annotation, coupled with the well-known issue of deep neural networks being susceptible to overfitting to randomly-labelled data (Zhang & Sabuncu, 2018), might lead to catastrophic failures, particularly in critical applications such as autonomous vehicles or medical diagnostics. These challenges have spurred research in the field of noisy label learning, aiming at addressing the problem of learning from datasets with noisy annotations.

More specifically, learning from noisy labels (LNL) aims to employ mis-labelled training data to train a classifier that can generalise well. One approach to accomplish this is to infer the clean label $Y$ of an instance $X$ from an observed noisy label $\hat{Y}$. If $C$ is the number of classes, then one can relate $Y$ and $\hat{Y}$ through the sum rule of probability as follows:

$$\Pr(\hat{Y}|X) = \sum_{c=1}^{C} \Pr(\hat{Y}|X, Y = c) \Pr(Y = c|X). \tag{1}$$

In the literature, the clean label probability $\Pr(Y|X)$, the transition probability $\Pr(\hat{Y}|X, Y)$ and the noisy label probability $\Pr(\hat{Y}|X)$ are modelled as categorical distributions. This, however, results in an ambiguous estimation because there are multiple combinations of hidden probability pair $(\Pr(\hat{Y}|X, Y), \Pr(Y|X))$ leading to the same observed noisy label probability $\Pr(\hat{Y}|X)$. For example, the following 3-class classification has at least two different solutions that result in the same

observed noisy label distribution $\Pr(\hat{Y}|X)$:

$$\Pr\left(\hat{Y}|X\right) = \begin{bmatrix} 0.25 \\ 0.45 \\ 0.3 \end{bmatrix} = \underbrace{\begin{bmatrix} 0.8 & 0.1 & 0.2 \\ 0.15 & 0.6 & 0.3 \\ 0.05 & 0.3 & 0.5 \end{bmatrix}}_{\Pr_1(\hat{Y}|X,Y)} \underbrace{\begin{bmatrix} 2/11 \\ 13/22 \\ 5/22 \end{bmatrix}}_{\Pr_1(Y|X)} = \underbrace{\begin{bmatrix} 0.7 & 0.2 & 0.1 \\ 0.1 & 0.6 & 0.1 \\ 0.2 & 0.2 & 0.8 \end{bmatrix}}_{\Pr_2(\hat{Y}|X,Y)} \underbrace{\begin{bmatrix} 2/15 \\ 7/10 \\ 1/6 \end{bmatrix}}_{\Pr_2(Y|X)}.$$

Failing to address the LNL identifiability issue might lead to estimating an undesirable model which may not be useful for making predictions or drawing conclusions. Existing LNL methods address the identifiability issue primarily through modelling-driven approaches, often by imposing ad-hoc priors in the form of heuristic assumptions and constraints, such as small-loss criterion (Han et al., 2018b), anchor points (Liu & Tao, 2015) or zero Bayes risk (Zhu et al., 2024)), feature-based methods (Kim et al., 2021) or unique constraints on the modelling of transition matrix (Li et al., 2021; Zhang et al., 2021b). Other methods follow a data-driven approach with multiple noisy labels per sample (Liu et al., 2023). Although methods based on the modelling-driven approach have achieved successful results in several benchmarks, their heuristics are either *(i)* sub-optimal, e.g., small-loss criterion might under-select "hard examples" (i.e., samples that are near decision boundaries) that are informative for learning, or *(ii)* associated with assumptions that might not always hold in practice, e.g., anchor points in (Liu & Tao, 2015), zero-error data in (Zhu et al., 2024), or "alignment clusterability" in (Kim et al., 2021). In contrast, the data-driven approach tackles the problem more fundamentally without relying on heuristic assumptions; however, current results in methods based on the data-driven approach are difficult to interpret (e.g., full-rank transition matrices (Liu et al., 2023)), reducing their applicability in practice.

In this paper, we investigate the identifiability issue in LNL following the data-driven approach with multiple noisy labels per training sample. Specifically,

- we formulate clean labels in LNL through multinomial mixture models and theoretically derive the identifiability condition, showing that at least $2C - 1$ i.i.d. noisy labels are required per training sample, and
- we propose a practical method based on nearest neighbours to generate additional i.i.d. noisy label to meet the identifiability condition.

The empirical evaluation of the proposed method evaluated on several LNL benchmarks (including synthetic, web-controlled and real-world label noise problems) demonstrates its capability to estimate clean labels without any heuristics. Furthermore, even though the main goal of this paper is the theoretical investigation of the identifiability condition, our practical method shows competitive results with several state-of-the-art techniques.

## 2 BACKGROUND

**Identifiability** studies whether the exact parameters of a model of interest can be uniquely recovered from observed data. Generally, a model is identifiable if and only if its parameters can be uniquely determined from available data. In contrast, a non-identifiable model implies that there exists multiple sets of parameters, where each set can explain the observed data equally well. Identifiability is particularly important in statistical modelling, where statistical methods are used to infer the true set of parameters from data. Formally, the identifiability of a distribution $\Pr(X;\theta)$ over a random variable $X$, parameterised by $\theta \in \Theta$ with $\Theta$ denoting a parametric space, can be defined as:

**Definition 1.** $\Pr(X;\theta)$ *is identifiable if it satisfies:* $\Pr(X;\theta) = \Pr(X;\theta') \implies \theta = \theta', \forall \theta, \theta' \in \Theta$.

**Mixture models** A mixture of $P$ distributions can be written as: $q(X) = \sum_{c=1}^{P} \pi_c \Pr(X;\theta_c)$, where $X$ is a random variable in $\mathcal{X}$, $\pi$ is the mixture coefficient vector in the $(P-1)$-dimensional probability simplex $\Delta^{P-1} = \left\{ \mathbf{y} : \mathbf{y} \in [0,1]^P \wedge \mathbf{1}^\top \mathbf{y} = 1 \right\}$, and $\{\Pr(X;\theta_c)\}_{c=1}^{P}$ is a set of $P$ distributions. Compared to a single distribution, mixture models are more flexible with higher modelling capacity, and hence, widely used to provide computationally convenient representation of complex data distributions. And since mixture models are an instance of latent variable models, the Expectation - Maximisation (EM) algorithm (Dempster et al., 1977) can be used to infer their parameters.

**Identifiability issue in mixture models** is one of the most common problems in statistical inference. For example, if all of the $P$ component distributions in a mixture model $q(X)$ belong

to the same parametric family, then $q(X)$ is invariant under $P!$ permutations by simply swapping the indices of the component distributions, a phenomenon known as *label-switching*. In practice, the identifiability issue due to *label-switching* (we will refer to this identifiability issue as label-switching from now on) is of no concern since one can impose an appropriate constraint on its parameters to obtain a unique solution. Nevertheless, parameter identifiability up to the permutation of class labels (we will refer this as identifiability in the remaining of this paper) is still a practical problem, at least in maximum likelihood for mixture models where the distribution components of such mixtures belong to certain distribution families. According to (Titterington et al., 1985, Section 3.1), most mixture models supported on continuous space, e.g., Gaussian mixture models (excluding the mixture of uniform distributions), are identifiable. However, when the support space is discrete, the identifiability of such mixtures might not always hold. For example, a mixture of Poisson distribution (Teicher, 1961) or a mixture of negative binomial distribution (Yakowitz & Spragins, 1968) is identifiable, while a mixture of binomial distributions is only identifiable under certain conditions (Teicher, 1961, Proposition 4). Another example is multinomial mixture models which is, according to the Theorem 1 defined below, identifiable if and only if the number of samples is at least almost twice the number of mixture components.

**Theorem 1** ((Kim, 1984, Lemma 2.2), (Elmore & Wang, 2003, Theorem 4.2)). *The class of $N$-trial $C$-category multinomial mixture models:* $\left\{ M(\mathbf{x}) : M(\mathbf{x}) = \sum_{c=1}^{C} \pi_c \operatorname{Mult}(\mathbf{x}; N, \rho_c) \right\}$ *is identifiable (up to label permutation) if and only if $N \geq 2C - 1$.*

## 3 METHODOLOGY

The first part of this section establishes the data-driven identifiability condition for the LNL problem in the setting where each training sample has multiple i.i.d. noisy labels. The second part introduces a practical method that satisfies the identifiability condition, enabling inference of the clean label distribution when only a single noisy label is available per training sample.

### 3.1 IDENTIFIABLE CONDITION FOR NOISY LABEL LEARNING

In LNL, the clean label $Y$ is often considered as a latent variable, and hence, the noisy label distribution $\Pr(\hat{Y}|X = \mathbf{x}_i)$ can be modelled as a mixture of $C$ distributions $\Pr(\hat{Y}|Y = c, X = \mathbf{x}_i), \forall c \in \{1, \ldots, C\}$ as in Eq. (1). Conventionally, each of the $C$ distributions $\Pr(\hat{Y}|Y = c, X = \mathbf{x}_i)$ is assumed to be a categorical distribution. Here, we expand the capability of such a modelling to the setting of multiple noisy labels by considering each $\Pr(\hat{Y}|Y = c, X = \mathbf{x}_i)$ as a multinomial distributions. Specifically, we model $\Pr(\hat{Y}|Y = c, X = \mathbf{x}_i) = \operatorname{Mult}(\hat{Y}; N, \rho_{ic})$ as an $N$-trial multinomial component, and $\Pr(Y = c|X = \mathbf{x}_i) = \pi_{ic}$ as the corresponding mixture coefficient, where $N \in \mathbb{Z}_+$ is the number of trials in the multinomial components (or number of noisy labels per training sample), $\rho_{ic} \in \Delta^{C-1}$ is the probability parameter of the multinomial component, $\pi_i \in \Delta^{C-1}$ is the clean label probability and $c \in \{1, \ldots, C\}$ is the class index. Eq. (1) can, therefore, be written in the form of a multinomial mixture model as:

$$\Pr\left(\hat{Y} \,\middle|\, X = \mathbf{x}_i\right) = \sum_{c=1}^{C} \pi_{ic} \operatorname{Mult}(\hat{Y}; N, \rho_{ic}). \tag{2}$$

The modelling assumption in Eq. (2) allows to determine the identifiable condition in LNL by leveraging the result in Theorem 1 as follows:

**Identifiability Condition** (Corollary of Theorem 1). *Any noisy label learning problem where the noisy label distribution is modelled as a multinomial mixture model shown in Eq. (2) is identifiable if and only if there are at least $2C - 1$ i.i.d. noisy labels $\hat{Y}$ of an instance $\mathbf{x}$ sampled from the noisy label multinomial distribution $\Pr\left(\hat{Y}|X = \mathbf{x}\right)$. In other words, $N \geq 2C - 1$.*

For example, the conventional LNL setting has only one noisy label per sample: $N = 1$, and hence, is non-identifiable for $C \geq 2$ unless additional assumptions or constraints are introduced. Another example is that binary classification on noisy labels, corresponding to $C = 2$, is identifiable if $N \geq 3$, or in other words, there must be at least 3 noisy labels per sample. This agrees with previous studies identifiability for the LNL problem (Zhu et al., 2021b; Liu et al., 2023).

Table 1: Comparison of identifiability conditions: Liu et al. (2023) vs. our approach.

| Aspect | (Liu et al., 2023) | Ours |
|---|---|---|
| *Assumption* | Full-rank transition matrix annotators $\Leftrightarrow$ experts whose annotations have high probability of matching ground truths | i.i.d. noisy labels from $\Pr\left(\hat{Y}\vert X\right)$ |
| *Min. № of labels per instance* | 3 expert annotations | $2C - 1$ i.i.d. annotations |
| *Pros & cons* | ✓ Few annotations per instance ✗ Expert recruitment is costly and hard to meet for large number of classes $C$ → prioritise annotator quality to minimise annotation quantity | ✗ Higher number of annotations ✓ Easy to achieve (e.g., crowdsourcing) → trade annotator quality for annotation quantity |

**Remark 1.** *Previous work by Liu et al. (2023) establishes an identifiability condition that requires at least three noisy labels per instance, each provided by a highly skilled annotator whose associated transition matrix is full rank. This assumption, while theoretically appealing, imposes a significant practical burden: recruiting such highly competent experts is costly and becomes increasingly infeasible as the number of classes $C$ grows. In contrast, our result "transposes" the previous condition, in which at least $2C - 1$ i.i.d. noisy labels from the same noisy label distribution $\Pr(\hat{Y}\vert X)$ are required per instance, without any requirement on annotator expertise beyond independence. Although our condition is less optimistic in terms of the number of labels needed, it eliminates the stringent high-skill (i.e., full rank) assumption. Intuitively, Liu et al. (2023) relies on a small set of expert annotators, whereas our result trades annotator quality for quantity, enabling the use of large-scale crowdsourcing. The differences between these two studies are summarised in Table 1.*

According to the identifiability condition, at least additional $2C - 2$ i.i.d. noisy labels per training sample must be available to solve the LNL in its standard setting. One can naively request more noisy labels per training sample, e.g., via crowd-sourcing, that satisfies the identifiability condition. Such an approach is, however, costly, time-consuming and poorly scalable, especially when the number of classes $C$ is large. For example, WebVision dataset (Li et al., 2017) with $C = 1,000$ classes will require at least an addition of 1,998 noisy labels per sample, resulting in an impractical solution. To address this issue, we propose a practical method in Section 3.2 to generate additional noisy labels to address the identifiability issue in LNL.

## 3.2 PRACTICAL METHOD TO FULFIL THE IDENTIFIABILITY CONDITION

To obtain additional noisy labels per sample without additional labelling resources, we propose to approximate the noisy label distribution $\Pr(\hat{Y}\vert X)$ by taking the similarity between the features of training samples into account. Our assumption is that training samples with similar features tend to be annotated similarly. In other words, similar instances have similar noisy labels (Table 5 empirically verifies this claim on Cifar-10N). Thus, we can leverage the single noisy label per training sample available in the training dataset to approximate the noisy label distribution $\Pr(\hat{Y}\vert X)$. The approximated distribution is then used to generate many i.i.d. noisy labels that meet the identifiability condition specified in Section 3.1. Subsequently, the EM algorithm is employed to infer the parameters of the multinomial mixture model in Eq. (2), including the clean label distribution $\Pr(Y\vert X)$. Appendix C presents a discussion on alternative ways to approximate $\Pr(\hat{Y}\vert X)$.

### 3.2.1 APPROXIMATING THE MULTI-MODAL NOISY LABEL DISTRIBUTION $\Pr(\hat{Y}\vert X)$

To generate additional i.i.d. noisy labels, we approximate the noisy label distribution of each training sample by exploiting the information of its nearest neighbours. The approximated noisy label distribution of an instance, denoted as $\widetilde{\Pr}(\hat{Y}\vert X = \mathbf{x}_i)$, is derived not only from its own noisy label but also from the noisy labels of other instances whose features are similar to the instance:

$$\widetilde{\Pr}(\hat{Y}\vert X = \mathbf{x}_i) \leftarrow \mu\, \widetilde{\Pr}(\hat{Y}\vert X = \mathbf{x}_i) + (1 - \mu) \sum_{j\neq i, j=1}^{K} \mathbf{A}_{ij}\, \widetilde{\Pr}(\hat{Y}\vert X = \mathbf{x}_j), \qquad (3)$$

where $\mu$ is a hyper-parameter in $[0, 1]$ reflecting the trade-off between the noisy labels of the instance and its neighbours, $K$ is the number of nearest neighbours, and $\mathbf{A}_{ij} \in [0, 1]$ is a coefficient representing the similarity between $\mathbf{x}_i$ and $\mathbf{x}_j$. Note that $\sum_{j \neq i, j=1}^{K} \mathbf{A}_{ij} = 1$.

There are several ways to find the similarity matrix $[\mathbf{A}_{ij}], \mathbf{A}_{ii} = 0, i \in \{1, \ldots, M\}, j \in \{1, \ldots, K\}$. For example, the study in (He et al., 2017) employs sparse subspace clustering method (Elhamifar & Vidal, 2013) to approximate the label distribution when learning human age from images. In this paper, we use a slightly similar but more efficient method that utilises the nearest neighbour information: locality-constrained linear coding (LLC) (Wang et al., 2010). In particular, the coefficient $\mathbf{A}_{ij}$ can be determined via the following optimisation:

$$\min_{\mathbf{A}_i} \|\mathbf{x}_i - \mathbf{B}_i \mathbf{A}_i\|_2^2 + \lambda \|\mathbf{d}_i \odot \mathbf{A}_i\|_2^2 \quad \text{s.t.:} \ \mathbf{1}^\top \mathbf{A}_i = 1, \mathbf{A}_{ij} \geq 0, \forall j \in \{1, \ldots, K\}, \quad (4)$$

where $\mathbf{B}_i$ is a matrix containing the $K$ nearest neighbours of instance $\mathbf{x}_i$ (each column is a nearest-neighbour instance), $\mathbf{A}_i = [\mathbf{A}_{i1} \quad \mathbf{A}_{i2} \quad \ldots \quad \mathbf{A}_{iK}]^\top$ is the $K$-dimensional vector representing the coding coefficients, $\odot$ is the element-wise multiplication (a.k.a. Hadamard product), $\mathbf{d}_i = \exp(\mathrm{dist}(\mathbf{x}_i, \mathbf{B}_i)/\sigma)$ is the locality adaptor with $\mathrm{dist}(\mathbf{x}_i, \mathbf{B}_i)$ being a vector of Euclidean distances from $\mathbf{x}_i$ to each of its nearest neighbours, and $\sigma$ being used for adjusting the weight decay speed for the locality adaptor. Nevertheless, since our interest is locality, not sparsity, in our implementation, we ignore the second term in Eq. (4) by setting $\lambda = 0$.

Note that the optimisation in (4) is slightly different from the original LLC due to the additional constraint of non-negativity of $\mathbf{A}_{ij}$. Nevertheless, the optimisation resembles a quadratic program, and therefore, can be efficiently solved by off-the-shelf solvers, such as OSQP (Stellato et al., 2020).

To efficiently find nearest neighbours, we utilise TPU-KNN (Chern et al., 2022) – an efficient approximation to search for nearest neighbours with GPU acceleration capabilities. To optimise computational efficiency and memory usage, we employ the features extracted from training samples in the nearest neighbour search. Furthermore, to enhance the scalability of our method when dealing with large datasets containing millions of training samples, we perform the nearest neighbour search in a subset (about 15,000 training samples) that is randomly sampled from the training set.

**Validity of i.i.d. assumption in generated noisy labels** The i.i.d. assumption in the identifiability condition is applied on noisy labels of training samples. It does not have any requirements on the estimation of the noisy label distribution $\Pr\left(\hat{Y}|X\right)$ or the dependence between neighbouring samples. Once this distribution is estimated via Eq. (3), noisy labels are i.i.d. sampled from the approximated distribution, and hence, satisfy the identifiability condition.

### 3.2.2 INFER CLEAN LABEL POSTERIOR WITH EM

Once the noisy label distribution $\widetilde{\Pr}(\hat{Y}|X)$ is approximated as a $(K+1)C$-multinomial mixture, we can generate $L$ sets, each consisting of $N$ noisy labels, with $N \geq 2C - 1$, for each instance. The EM algorithm is then used to infer the parameter of the multinomial mixture model in Eq. (2). In particular, the objective function for the $i$-th sample can be written as:

$$\max_{\pi_i, \rho_i} \frac{1}{L} \sum_{l=1}^{L} \ln \Pr(\hat{Y} = \hat{\mathbf{y}}_l | X = \mathbf{x}_i; \pi_i, \rho_i) + \ln \Pr(\pi_i; \alpha) + \ln \Pr(\rho_i; \beta), \quad (5)$$

where: $\hat{\mathbf{y}}_l \sim \widetilde{\Pr}(\hat{Y}|X = \mathbf{x}_i)$ is an $N$-trial multinomial vector (e.g., sum of $N$ one-hot noisy labels of an instance), and $\alpha$ and $\beta$ are the parameters of the priors of $\pi_i$ and $\rho_i$, respectively. The parameters $\pi_i$ and $\rho_i$ in (5) can be optimised via the EM algorithm.

According to Eq. (3), additional multinomial noisy labels are sampled from a $(K+1)C$-multinomial mixture, $\widetilde{\Pr}(\hat{Y}|X = \mathbf{x}_i)$. Such a sampling process, however, has a complexity of $\mathcal{O}((K+1)C^2)$, which is expensive when $C$ – the number of classes – is large. That is because the mixture coefficient (or pseudo-clean label probability), $\pi_i = \Pr(Y|X = \mathbf{x}_i)$, is assumed to be dense with $C$ components, while in practice, $\Pr(Y|X = \mathbf{x}_i)$ is often sparse with only $C_0$ components where $C_0 \ll C$ (Han et al., 2018a). We therefore exploit this observation to mitigate the issue of high complexity due to sampling. Appendix F provides further details on the reduction of number of noisy labels needed in our practical implementation.

The proposed method (see Algorithm 1 in Appendix D) relies on the extracted features to perform nearest neighbour search. Thus, if the features extracted are biased, it will worsen the quality of the

Table 2: The running time complexity per epoch of the data pre-processing step of the proposed algorithm and existing methods, where: $|\theta|$ is the number of model's parameters, $M$ is the total number of training samples, $B$ is the mini-batch size, $C$ is the number of classes, $K$ is the number of nearest neighbours, L is the set of multiple noisy labels (e.g., $2C-1$ per training samples), $d$ is the dimension of input samples, $n_{\text{augment}}$ is the number of data augmentations, $n_{\text{iter}}, n_{\text{osqp}}, n_{\text{em}}$ are the number of optimisation iterations used within each method.

| Method | Complexity |
|---|---|
| DivideMix (Li et al., 2020) | $\mathcal{O}\left(6|\theta| + \left[4 + {}^2\!/_B \left(n_{\text{augment}}d + 2C\right)\right] M\right)$ |
| HOC (Zhu et al., 2021b) | $\mathcal{O}\left(|\theta| + 3M + 2\ln M + n_{\text{iter}}C^2\right)$ |
| **Ours** | $\mathcal{O}\left(2|\theta| + 2\ln M + 2n_{\text{osqp}}Kd + 2(L + n_{\text{em}})C^2\right)$ |

nearest neighbours, reducing the effectiveness of the proposed method. To avoid such confirmation bias, we follow the *co-teaching* approach (Han et al., 2018b) that trains two models simultaneously where the noisy labels being cleaned by one model are used to train the other model and vice versa. We also analyse the complexity (only the "data pre-processing step" and excludes the loss calculation and model training because they are almost identical) of the proposed algorithm (see Algorithm 1) and present the result in Table 2 (see Appendix E for the detailed analysis). In general, the bottleneck of our method is at the sampling of i.i.d. noisy labels and the EM algorithm due to its quadratic complexity with respect to the number of classes $C$. Readers are referred to Table 9 in Appendix E for the details of actual running time.

## 4 EXPERIMENTS

We employ several LNL benchmarks to evaluate the robustness of the proposed learning method when dealing with the most realistic type of label noise, namely: the instance-dependent noise. In particular, the experiments are performed on both synthetic and real-world instance-dependent label noise benchmarks. In addition, because our focus is on the theory of the identifiability in the LNL problem, we show that the proposed method is effective and competitive to other state-of-the-art (SOTA) methods without resorting to fine-tuning or employing highly-complex neural network architectures. The details of datasets, hyper-parameters and models used are shown in Appendix G.

### 4.1 COMPARISON WITH IDENTIFIABILITY-BASED METHODS

Since both (Liu et al., 2023) and our study tackle the same identifiability issue in LNL, but follow different approaches, it is important to evaluate the performance of practical methods derived from the two approaches. More specifically, we compare HOC (Zhu et al., 2021b), representing (Liu et al., 2023), with our method presented in Section 3.2. The comparison is conducted on multiple noisy-label datasets, namely: three human-annotated noisy labels in CIFAR-10N (Wei et al., 2022). The results of HOC are obtained through its official implementation, which is publicly available.

For the real-world dataset CIFAR-10N, we evaluate on all of the available settings, including a single label for each of the three annotation cases, the *aggregate* which randomly selects one label from the three noisy labels, and the *worst* which selects the noisy label among the three labels annotated. We also consider the case of combining three noisy labels together by aggregating them into a soft label in the case of the cross-entropy baseline and our method, or passing all three into the model of interest to learn higher-order statistics in the case of HOC.

As shown in Table 3, our method outperforms the cross-entropy baseline and HOC in all CIFAR-10N settings. The performance gap between HOC and our method may be attributed to HOC's dependence on *k-NN label clusterability* (Zhu et al., 2021b, Definition 1), which requires that the k-nearest neighbours of an instance must belong to the same true class. This is evident from the improvement of HOC's performance when using three noisy labels per training sample, as shown in the last column of Table 3. In contrast, our method does not rely on the strong assumption of k-NN label clusterability and can consistently perform well with either single or multiple noisy labels per training sample. Note that when using all three available noisy labels, the performance gap between the baseline (training model directly on noisy label data), HOC and our method vanishes. This might be because the assumption of three noisy labels in HOC becomes valid. In addition, the

Table 3: Prediction accuracy on human-annotation CIFAR-10N.

| Setting | CIFAR-10N | | | | | |
| | Aggregate | Random 1 | Random 2 | Random 3 | Worst | 3 noisy labels |
| Noise rate | 0.09 | 0.17 | 0.18 | 0.18 | 0.40 | 0.02 |
|---|---|---|---|---|---|---|
| Cross-entropy (Wei et al., 2022) | $87.77 \pm 0.38$ | $85.02 \pm 0.65$ | $86.46 \pm 1.79$ | $85.16 \pm 0.61$ | $77.69 \pm 1.55$ | $92.24 \pm 0.66$ |
| HOC (Zhu et al., 2021b) | $83.34 \pm 0.09$ | $81.92 \pm 0.18$ | $81.76 \pm 0.12$ | $81.31 \pm 0.17$ | $62.31 \pm 0.14$ | $91.94 \pm 0.73$ |
| **Ours** | $\mathbf{89.69} \pm 1.15$ | $\mathbf{90.00} \pm 0.23$ | $\mathbf{89.79} \pm 0.18$ | $\mathbf{88.69} \pm 0.23$ | $\mathbf{89.89} \pm 0.45$ | $\mathbf{92.41} \pm 0.79$ |

Table 4: Prediction accuracy (%) on real-world noisy label datasets: *(left)* Red CNWL, *(middle)* mini-WebVision and ImageNet and *(right)* Animal-10N. Best result in **bold**, $2^{nd}$ best in *italics*.

| Method | Noise rate of CNWL | | |
| (no pre-trained) | 0.2 | 0.4 | 0.6 |
|---|---|---|---|
| Cross-entropy | 47.36 | 42.70 | 37.30 |
| mixup | 49.10 | 46.40 | 40.58 |
| DivideMix | 50.96 | 46.72 | 43.14 |
| MentorMix | 51.02 | 47.14 | 43.80 |
| FaMUS | 51.42 | 48.06 | 45.10 |
| SSR* | 52.18 | 48.96 | 42.42 |
| LSL | **54.68** | **49.80** | *45.46* |
| **Ours** | *52.78* | *49.18* | **46.00** |

| Method | WebVision | ImageNet |
|---|---|---|
| mixup | 74.96 | - |
| Co-teaching | 63.58 | 61.48 |
| DivideMix | 77.32 | **75.20** |
| ELR | 76.26 | 68.71 |
| MOIT | *78.36* | - |
| NCR | 77.10 | - |
| ASL | 66.68 | 64.12 |
| ROBOT | 68.24 | 65.20 |
| PCSE | 70.48 | 67.72 |
| **Ours** | **80.48** | *74.63* |

| Method | Animal-10N |
|---|---|
| Cross entropy | 79.40 |
| Nested-Dropout | 81.30 |
| CE + Dropout | 81.30 |
| SELFIE | 81.80 |
| PLC | 83.40 |
| Nested-CE | *84.10* |
| ASL | 77.70 |
| ROBOT | 83.52 |
| PCSE | 83.82 |
| **Ours** | **85.96** |

label noise rate in this case is too small (approximately 0.02), and hence, makes the comparison less distinguishable. Note that in this experiment, our method relies on a PreAct Resnet-18 pre-trained on the training set of CNWL using SimCLR (Chen et al., 2020) for 500 epochs with a similar data augmentation policy (random crop and resize, colour jittering, grey or colourise and Gaussian blur), while the nearest neighbours in HOC rely on a Resnet-34 pre-trained on ImageNet.

## 4.2 RESULTS ON LNL COMMON BENCHMARKS

For Red CNWL, we follow the experimental setup from (Xu et al., 2021) and present results in Table 4 *(left)*. This benchmark includes widely used SOTA methods evaluated on low-resolution (32-by-32 pixel$^2$) images to ensure a fair comparison. While our goal is not to outperform existing methods, these results illustrate that generating multiple noisy labels per sample shows competitive performance, supporting our theoretical claims. We further evaluate the method on real-world noisy label datasets, mini-WebVision and Animal-10N, with results shown in Table 4 *(middle)* and *(right)*. For mini-WebVision, we initialise the model with a ResNet-50 pre-trained for 100 epochs using DINO (Caron et al., 2021). For Animal-10N, we use a VGG-19 pre-trained with DINO for 800 epochs. Again, the results are not intended to be SOTA but to show that the proposed approach is robust and performs comparably under realistic noisy conditions. Additional results on CIFAR-10 and CIFAR-100 are provided in Appendix H (see Table 10). Table 10 *(top)* compares our method to other approaches on synthetic noise settings. On CIFAR-10, our method performs on par with SOTA methods, and on CIFAR-100, it slightly outperforms them. These results further support the core claim: that leveraging a sufficient number of noisy labels per sample can effectively address the noisy-label problem.

## 4.3 ABLATION STUDIES

We further study the effect of the number of noisy labels per sample, the number of nearest neighbours and the effectiveness of our relabelling. Note that no self-supervised learning is used for pre-training the model in the ablation studies to avoid potential confounding factors.

**Number of noisy labels per sample** We run experiments on the 100-class LNL problem of the Red CNWL dataset at 0.6 noise rate with various number of noisy labels per sample $N \in \{3, 20, 100, 199, 400\}$. We plot the results in Fig. 1 *(left)*, where $L$ is the number of $N$ multinomial noisy labels defined in Algorithm 1. When $L$ is small, the more noisy labels per sample or larger $N$, the more effective, and the effectiveness diminishes after the threshold of $2C - 1$, which in this case is 199. This empirically confirms the validity of Section 3.1 about the identifiability in noisy label learning. However, when $L$ is large, the performance difference when varying $N$ is not as noticeable. In this regime (of large $L$), Section 3.1 might result in a conservative requirement in terms of number of noisy labels per sample. The current LNL setting might contain some common

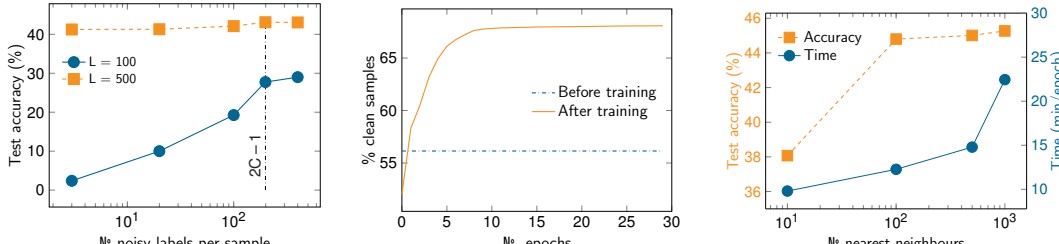

Figure 1: Ablation studies on: *(left)* the effect of number of noisy labels per sample on Red CNWL at a noise rate of 0.6, *(middle)* and *(right)* the accuracy of the relabelling and the influence of nearest neighbours on CIFAR-100.

Table 5: Averaged label agreement (in percentage) of $K$ nearest neighbours on CIFAR-10N to verify the assumption of label consistency.

| Settings | K = 10 | | K = 100 | |
|---|---|---|---|---|
| | **Train until converged** | **Warmup** | **Train until converged** | **Warmup** |
| 3 labels (2% noise) | 97.96 | 65.49 | 62.10 | 55.25 |
| Random 1 (17% noise) | 89.04 | 60.95 | 54.29 | 42.52 |
| Random 2 (18 % noise) | 88.68 | 59.70 | 54.36 | 42.58 |
| Random 3 (18% noise) | 88.79 | 59.73 | 54.79 | 42.48 |
| Worst label (40% noise) | 53.73 | 45.05 | 43.15 | 33.65 |

latent structure between samples (e.g., limited number of candidate labels per instance), which we have not exploited yet to bring down the number of required noisy labels per sample. Future work will need to address such issue to make the problem more practical.

**Effectiveness of label cleaning** is investigated by measuring the accuracy on the training set between the pseudo labels "cleaned" by EM and the ground truth labels on CIFAR-100 at a noise rate of 0.5. We also include the percentage of clean samples before training to compare more easily. Note that despite the nominal noise rate of 0.5, the "empirical" noise rate measured on the generated noisy labels following (Xia et al., 2020) before training is 0.44 (corresponding to 56 percent clean samples). The results in Fig. 1 *(middle)* show that the proposed method can improve from the initial dataset with 56 percent of clean samples to 68 percent. This 12 percent improvement is equivalent to cleaning 27 percent of noisy labels that are initially present in the training set.

**Number of nearest neighbours** We also investigate the effect of the number of nearest neighbours $K$ to our proposed method by evaluating on CIFAR-100 at a noise rate of 0.5. The results in Fig. 1 *(right)* show that the larger $K$, the more accurate the testing accuracy. However, the trade-off is the running time shown in the right axis of Fig. 1 *(right)*. Since $K = 100$ gives a good balance between the performance and running time, this value is used in all of our experiments.

**Label consistency of KNN** We verify the label consistency of nearest neighbours with features extracted from our models. It is measured as the average label agreement between the sample of interest and its nearest neighbours. Table 5 shows the label agreement evaluated at two checkpoints: (i) training the model directly on noisy labels until convergence (*train directly*), and (ii) training the model for only 10 epochs (*warm-up*) In general, approximately more than two thirds of the nearest neighbours of each sample share the same class label at the beginning of the training. This means that the assumption of label consistency, in which similar instances have similar noisy labels, made in Section 3.2.1 holds with adequate probability. Additional results on class-dependent (or asymmetric) label noise are included in Appendix I.

## 5 RELATED WORK

LNL has been studied since the 1980s with some early works focusing on the statistical point of view (Angluin & Laird, 1988; Bshouty et al., 2002), such as determining the number of samples to achieve certain prediction accuracy under certain types of label noise. The field has then attracted more research interest, especially in the era of deep learning where an increasing number of anno-

tated data is required to train large deep learning models. Learning from noisy labels is inherently challenging due to the issue of identifiability. Despite its importance, the identifiability issue in LNL remains a partially-addressed problem that requires the introduction of model-driven ad-hoc constraints or exploration through the use of data-driven multiple noisy labels.

**Ad-hoc constraints** Many studies have implicitly or partially discussed the identifiability issue in the LNL problem and proposed practical methods designed with different heuristic criteria (Menon et al., 2015) to make the problem identifiable. The most widely-used constraint is the *small loss criterion* where the labels of samples with small loss values are assumed to be clean (Han et al., 2018b). Training is then carried out either on only those low-risk samples (Han et al., 2018b) or cast as a semi-supervised learning approach with those clean samples representing labelled data while the others denoting un-labelled data (Li et al., 2020). Although this line of research achieves remarkable results in several benchmarks, they still lack theoretical foundations that explain why the *small loss criterion* is effective. There is one recent attempt that theoretically investigates the *small loss hypothesis* (Gui et al., 2021), but the study is applicable only to the class-dependent (a.k.a. instance-independent) label noise setting that assumes the presence of "anchor points", i.e., samples that are guaranteed to have clean labels. Other methods propose different ad-hoc constraints based on observations in matrix decomposition and geometry. For instance, Lin et al. (2015); Li et al. (2021) suggest the minimal volume of the simplex formed from the columns of transition matrices. Zhang et al. (2021b) present a matrix decomposition approach and employ total variation regularisation to ensure the uniqueness of the solution. Cheng et al. (2022) impose similarity of transition matrices between samples that are close to each other.

**Multiple noisy labels per instance** Learning from multiple noisy labels per instance has recently emerged as one approach to theoretically address the identifiability issue. The most relevant study in this area is the investigation of identifiability of transition matrices in noisy label learning (Liu et al., 2023). In that paper, the authors implicitly extend the conventional 2-D transition matrix to a 3-D tensor with the third dimension representing annotators and exploit the results in 3-D arrays (Kruskal, 1976; 1977) to find the condition of identifiability. Similar to a study in crowd-sourcing literature (Traganitis et al., 2018, Lemma 1), the authors in (Liu et al., 2023) conclude that at least 3 "informative" noisy labels per instance are needed. Although the finding is more optimistic than ours, it relies on the assumption of "informative" noisy labels, which requires a full-rank transition matrix for each annotator on each instance. The assumption of full-rank transition matrices implicitly depends on the number of classes as a larger number of classes increases the difficulty to make each $C$-by-$C$ transition matrix full-rank. Moreover, that assumption lacks clarity since it is unclear how to translate the full-rankness required for a transition matrix to a property an annotator must have. In contrast, our result does not rely on those assumptions, such as the "informativeness" of noisy labels nor full-ranked transition matrices, except that the multiple noisy labels should be i.i.d., and that we rely on a multinomial mixture modelling. Another related study is the Higher-Order-Consensus (HOC) (Zhu et al., 2021b), which is a practical method present in (Liu et al., 2023). To address the identifiability issue in LNL, HOC also relies on the full-rank transition matrices (Zhu et al., 2021b, Assumption 1) and the *2-NN label clusterability* (Zhu et al., 2021b, Definition 1) where the sample of interest and its two nearest neighbours belong to the same true class. HOC, however, mainly relies on instance-independent label noise, which may limit its applicability. Furthermore, the assumptions of clusterability in HOC does not always hold in practice (Zhu et al., 2021b, Table 3), especially at larger noise rates. Compared to HOC, our method presented in Section 3.2 requires a less restricted assumption where similar samples are annotated similarly (see Table 5 for our empirical verification). We also provide an empirical comparison between HOC and our practical method in Section 4.1 to understand further the differences.

## 6 CONCLUSION

This study has conducted a formal investigation into the identifiability of noisy label learning using multinomial mixture models. Specifically, the LNL problem has been formulated as a multinomial mixture model, where the clean label probability is represented as the mixture coefficient and each column in the transition matrix is represented as each multinomial component. Such modelling reveals that LNL is identifiable when there are at least $2C - 1$ i.i.d. noisy labels per sample provided; otherwise, the problem becomes non-identifiable unless additional assumptions or constraints are employed. This result agrees with previous studies on the identifiability of label noise learning, where the conventional setting of a single noisy label per training sample is non-identifiable. To

practically address the LNL problem, we propose to leverage nearest neighbours to generate additional noisy labels to fulfill the identifiability requirement. The clean label distribution is then inferred through the EM algorithm for multinomial mixture models. Even though our goal was not to outperform SOTA methods, the experimental results show that generating multiple noisy labels per sample yields competitive performance on various challenging benchmarks, particularly in scenarios involving instance-dependent and real-world label noises, supporting our theoretical claims. The proposed method also out-performs HOC – a practical method that deals with the identifiability in noisy label learning – in several settings, including the one with multiple noisy labels per training sample. Despite the promising finding, the number of noisy labels required to make the LNL identifiable in Section 3.1 is still impractical in several applications where the number of classes is large, if we require additional manual labels. Future work will focus on the relation between class labels to further reduce this number, making it more practical.

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

## A  DISCLOSURE OF LARGE LANGUAGE MODEL USAGE

Portions of the text in this paper were revised through the usage of large language models to improve clarity. The original idea, formulation, experiments and conclusion were done by the authors themselves.

## B  EM FOR MULTINOMIAL MIXTURE MODELS

This appendix presents the EM algorithm for multinomial mixture models mentioned in Section 3.2.2.

Recall that a mixture of multinomial distributions in the LNL problem can be written as:

$$p(\hat{Y}|X = \mathbf{x}_i) = \sum_{c=1}^{C} p(Y = c|X = \mathbf{x}_i)\, p(\hat{Y}|X = \mathbf{x}_i, Y = c) = \sum_{c=1}^{C} \pi_{ic}\text{Mult}(\hat{Y}; N, \rho_{ic}). \quad (2)$$

The aim here is to infer the parameters of the multinomial mixture model. In particular, we want to exploit the given $L$ noisy labels $\{\hat{y}_l\}_{l=1}^{L}$ of an instance $X = \mathbf{x}_i$ to infer $p(Y|X = \mathbf{x}_i)$ and $p(\hat{Y}|X = \mathbf{x}_i, Y)$.

We note that despite the identifiable condition in Section 3.1 requiring $N \geq 2C - 1$, we still need multiple sets of such $N$-trial noisy labels for inference. If only a single set of $N$ noisy labels is given, the inference will have a very large uncertainty although the problem is identifiable.

### B.1  MAXIMUM LIKELIHOOD

Given $L$ noisy labels $\{\hat{y}_l\}_{l=1}^{L}$ of an instance $X = \mathbf{x}_i$, the objective in terms of maximum likelihood estimation can be written as:

$$\max_{\pi_i, \rho_i} \sum_{l=1}^{L} \ln p(\hat{Y} = \mathbf{y}_l|X = \mathbf{x}_i) = \max_{\pi_i, \rho_i} \sum_{l=1}^{L} \sum_{c=1}^{C} \pi_{ic}\text{Mult}(\hat{Y} = \mathbf{y}_l; N, \rho_{ic}). \quad (6)$$

#### B.1.1  E-STEP

This step is to calculate the posterior of the latent variable $\mathbf{z}_n$ given the data $\mathbf{x}_n$:

$$\begin{aligned}
\gamma_{lc} &= p(Y = c|\hat{Y} = \hat{\mathbf{y}}_l, X = \mathbf{x}_i; \pi_i^{(t)}, \rho_i^{(t)}) \\
&= \frac{p(\hat{Y} = \hat{\mathbf{y}}_l|Y = c, X = \mathbf{x}_i; \rho_i^{(t)})\, p(Y = c|X = \mathbf{x}_i; \pi_i^{(t)})}{\sum_{c=1}^{C} p(\hat{Y} = \hat{\mathbf{y}}_l|Y = c, X = \mathbf{x}_i; \rho_i^{(t)})\, p(Y = c|X = \mathbf{x}_i; \pi_i^{(t)})} \\
&= \frac{\pi_c^{(t)}\,\text{Mult}(\hat{Y} = \hat{\mathbf{y}}_l; N, \rho_{ic}^{(t)})}{\sum_{c=1}^{C} \pi_{ic}^{(t)}\,\text{Mult}(\hat{Y} = \hat{\mathbf{y}}_l; N, \rho_{ic}^{(t)})}.
\end{aligned} \quad (7)$$

#### B.1.2  M-STEP

In the M-step, we maximise the following expected completed log-likelihood w.r.t. $\pi_i$ and $\rho_i$:

$$\begin{aligned}
Q &= \sum_{l=1}^{L} \mathbb{E}_{p(Y|\hat{Y}=\hat{\mathbf{y}}_l, X=\mathbf{x}_i; \pi_i^{(t)}, \rho_i^{(t)})}\left[\ln p(\hat{Y}, Y|X = \mathbf{x}_i; \pi_i, \rho_i)\right] \\
&= \sum_{l=1}^{L} \mathbb{E}_{p(Y|\hat{Y}=\hat{\mathbf{y}}_l, X=\mathbf{x}_i; \pi_i^{(t)}, \rho_i^{(t)})}\left[\ln p(Y|X = \mathbf{x}_i; \pi_i) + \ln p(\hat{Y} = \hat{\mathbf{y}}_l|Y, X = \mathbf{x}_i; \rho_i)\right] \\
&= \sum_{l=1}^{L}\sum_{c=1}^{C} \mathbb{E}_{p(Y=c|\hat{Y}=\hat{\mathbf{y}}_l, X=\mathbf{x}_i; \pi_i^{(t)}, \rho_i^{(t)})}\left[\ln \pi_{ic} + \ln \text{Mult}(\hat{Y} = \hat{\mathbf{y}}_l; N, \rho_{ic})\right] \\
&= \sum_{l=1}^{L}\sum_{c=1}^{C} \gamma_{nk}\left[\ln \pi_{ic} + \sum_{c'=1}^{C} \hat{\mathbf{y}}_l \ln \rho_{icc'} + \text{const.}\right].
\end{aligned} \quad (8)$$

The Lagrangian for $\pi$ can be written as:

$$Q_{\pi_i} = Q - \lambda \left( \sum_{c=1}^{C} \pi_{ic} - 1 \right), \tag{9}$$

where $\lambda$ is the Lagrange multiplier. Taking derivative of the Lagrangian w.r.t. $\pi_{ic}$ gives:

$$\frac{\partial Q_{\pi_i}}{\partial \pi_{ic}} = \frac{1}{\pi_{ic}} \sum_{l=1}^{L} \gamma_{lc} - \lambda. \tag{10}$$

Setting the derivative to zero and solving for $\pi_{ic}$ gives:

$$\pi_{ic} = \frac{1}{\lambda} \sum_{l=1}^{L} \gamma_{lc}. \tag{11}$$

And since $\sum_{c=1}^{C} \pi_{ic} = 1$, one can substitute and find that $\lambda = L$. Thus:

$$\boxed{\pi_{ic}^{(t+1)} = \frac{1}{L} \sum_{l=1}^{L} \gamma_{lc}.} \tag{12}$$

Similarly, the Lagrangian of $\rho_{icc'}$ can be expressed as:

$$Q_{\rho_{icc'}} = Q - \sum_{c=1}^{C} \eta_c \left( \sum_{c'=1}^{C} \rho_{icc'} - 1 \right), \tag{13}$$

where $\eta_c$ is the Lagrange multiplier. Taking derivative w.r.t. $\rho_{icc'}$ gives:

$$\frac{\partial Q_{\rho_{icc'}}}{\partial \rho_{icc'}} = \frac{1}{\rho_{icc'}} \sum_{l=1}^{L} \gamma_{lc} \hat{\mathbf{y}}_{lc'} - \eta_c. \tag{14}$$

Setting the derivative to zero and solving for $\rho_{icc'}$ gives:

$$\rho_{icc'} = \frac{1}{\eta_c} \sum_{l=1}^{L} \gamma_{lc} \hat{\mathbf{y}}_{lc'}. \tag{15}$$

The constraint on $\rho_{ic}$ as a probability vector leads to $\eta_c = N \sum_{l=1}^{L} \gamma_{lc}$. Thus:

$$\boxed{\rho_{icc'}^{(t+1)} = \frac{\sum_{l=1}^{L} \gamma_{lc} \hat{\mathbf{y}}_{lc'}}{N \sum_{l=1}^{L} \gamma_{lc}}.} \tag{16}$$

## B.2 MAXIMUM A POSTERIOR (MAP)

The objective function is similar to the one in Appendix B.1, except including the prior on $\pi_i$ and $\rho_i$ as follows:

$$\max_{\pi_i, \rho_i} Q := \sum_{l=1}^{L} \sum_{c=1}^{C} \pi_{ic} \text{Mult}(\hat{Y} = \mathbf{y}_l; N, \rho_{ic}) + \ln p(\pi_i; \alpha) + \sum_{c=1}^{C} \ln p(\rho_{ic}; \beta), \tag{17}$$

where the two priors are:

$$p(\pi_i; \alpha) = \text{Dir}(\pi_i; \alpha) \tag{18}$$
$$p(\rho_{ic}; \beta) = \text{Dir}(\rho_{ic}; \beta). \tag{19}$$

The E-step in this case remains unchanged from (7).

The derivative of the Lagrangian for $\pi$ can be written as:

$$\frac{\partial Q_{\pi_i}^{\mathrm{MAP}}}{\partial \pi_{ic}} = \frac{1}{\pi_{ic}} \left( \sum_{l=1}^{L} \gamma_{lc} + \alpha_c - 1 \right) - \lambda. \tag{20}$$

Thus:

$$\boxed{\pi_{ic}^{(t+1)} = \frac{\sum_{l=1}^{L} \gamma_{lc} + \alpha_c - 1}{L + \sum_{c=1}^{C} \alpha_c - C}.} \tag{21}$$

Similarly for $\rho_{icc'}$:

$$\frac{\partial Q_{\rho}^{\mathrm{MAP}}}{\partial \rho_{icc'}} = \frac{1}{\rho_{icc'}} \left( \sum_{l=1}^{L} \gamma_{lc} \hat{\mathbf{y}}_{lc'} + \beta_{c'} - 1 \right) - \eta_c. \tag{22}$$

Thus:

$$\boxed{\rho_{icc'}^{(t+1)} = \frac{\sum_{l=1}^{L} \gamma_{lc} \hat{\mathbf{y}}_{lc'} + \beta_{c'} - 1}{N \sum_{l=1}^{L} \gamma_{lc} + \sum_{c'=1}^{C} \beta_{c'} - C}.} \tag{23}$$

## C   ALTERNATIVE WAYS TO APPROXIMATE THE NOISY LABEL DISTRIBUTION

There might be different ways to approximate $p(\hat{Y}|X)$, e.g., simply using a neural network trained directly on noisy-label data $\{(\mathbf{x}_i, \hat{y}_i)\}_{i=1}^{M}$. This approach is, however, sub-optimal since the noisy label distribution $p(\hat{Y}|X)$ is modelled as a simple categorical distribution (represented by the softmax output of the neural network). Inferring a mixture of multinomial distributions from samples generated from such a less expressive distribution may potentially result in a single-component mixture (i.e., component collapse), making the estimation inaccurate. Another way is to train a neural network to explicitly output $p(Y|X)$ and $p(\hat{Y}|X,Y)$ (Goldberger & Ben-Reuven, 2017), which is equivalent to learn a mixture of categorical distributions. This would, however, be subjected to the identifiability issue in LNL. In contrast, our approach exploits the noisy labels of nearest neighbours to approximate the noisy label distribution. In particular, we model the distribution of noisy label of each instance as a mixture of $C$ multinomial components, and hence, the approximated noisy label distribution obtained through $K$ nearest neighbours would be a mixture of $(K+1)C$ multinomial distributions (please refer to Eq. (3)). In addition, exploiting nearest neighbours in the feature space results in more consistent label distributions across samples (Iscen et al., 2022). Furthermore, the nearest-neighbour based approach has been demonstrated to be effective and widely-used in label distribution learning (He et al., 2017).

## D   PROPOSED LEARNING ALGORITHM TO OVERCOME THE IDENTIFIABILITY IN LABEL NOISE

The proposed learning algorithm to augment the noisy labels is shown in Algorithm 1.

## E   RUNNING TIME COMPLEXITY ANALYSIS

We analyse the complexity of our proposed method, in which the pseudo-clean labels are inferred from the noisy label distributions. Our analysis focuses on the "pre-processing" step right before calculating loss and back-propagation because this is the main difference between these methods (the loss calculation and gradient update for the model's parameters are similar). Hence, in the following analysis, we omit the complexity of relating to the loss calculation and parameter update.

Another note is that the complexity is analysed for one epoch. For the convenience, the notations used are explicitly defined in Table 6.

---

**Algorithm 1** Progressively clean noisy labels

---

1: **procedure** TRAIN($\mathbf{X}, \hat{\mathbf{Y}}, \mu, \eta\, \gamma$)
2:  ▷ $\mathbf{X} \in \mathbb{R}^{d \times M}$: *matrix of $M$ instances*            ◁
3:  ▷ $\hat{\mathbf{Y}} \in \mathbb{R}^{C \times M}$: *$M$ one-hot noisy labels*          ◁
4:  ▷ *$K$: № nearest neighbours*                ◁
5:  ▷ *$L$: № N-trial multinomial samples*           ◁
6:  ▷ *$\mu$: trade-off coefficient*               ◁
7:  ▷ *$\eta$: № EM iterations*                ◁
8:  ▷ *$\gamma$: a weighting factor to update multinomial mixture model's parameters* ◁
9:  initialise $\Pi = \{\pi_i : \pi_i \leftarrow \text{SOFT LABEL}(\hat{\mathbf{y}}_i)\}_{i=1}^{M}$
10: initialise $P = \{\rho_i : \rho_i \leftarrow \mathbf{I}_{C \times C}\}_{i=1}^{M}$     ▷ *Random diagonal-dominant matrices*
11: initialise feature extractor $\theta$ and a classifier $\mathbf{w}$
12: warm-up: $(\theta, \mathbf{w}) \leftarrow \text{TRAIN}(\mathbf{X}, \Pi, (\theta, \mathbf{w}))$
13: **while** $(\pi, \rho)$ not converged **do**
14:  $\Pi' \leftarrow \varnothing, P' \leftarrow \varnothing$       ▷ *store inferred parameters of $p(Y|X)$ and $p(\hat{Y}|X, Y)$*
15:  **for** each $\mathbf{x}_i \in \mathbf{X}$ **do**
16:   extract features: $f(\mathbf{x}_i; \theta)$
17:   find $K$ nearest neighbours: $\mathbf{B}_i \leftarrow \text{KNN}(f(\mathbf{x}_i; \theta))$
18:   calculate similarity matrix: $\mathbf{A}_i \leftarrow \text{LLC}(f(\mathbf{x}_i; \theta), \mathbf{B}_i)$     ▷ *Eq. (4)*
19:   approximate noisy label distribution $\tilde{p}(\hat{Y}|X = \mathbf{x}_i)$     ▷ *Eq. (3)*
20:   generate multinomial noisy labels: $\tilde{\mathbf{Y}}_i = \{\hat{\mathbf{y}}_l : \hat{\mathbf{y}}_l \sim \tilde{p}(\hat{Y}|X = \mathbf{x}_i)\}_{l=1}^{L}$
21:   infer mixture model parameters: $\pi_i', \rho_i' \leftarrow \text{EM}(\tilde{\mathbf{Y}}_i, \eta)$
22:   update clean label: $\pi_i \leftarrow \gamma\pi_i + (1 - \gamma)\pi_i'$
23:   update parameters of multinomial components: $\rho_i \leftarrow \gamma\rho_i + (1 - \gamma)\rho_i'$
24:   store clean label: $\Pi' \leftarrow \Pi' \cup \pi_i$
25:   store probability vectors of multinomial components: $P' \leftarrow P' \cup \rho_i$
26:  update parameters of clean labels and multinomial components: $\Pi \leftarrow \Pi', P \leftarrow P'$
27:  train model: $(\theta, \mathbf{w}) \leftarrow \text{TRAIN}(\mathbf{X}, \Pi, (\theta, \mathbf{w}))$
28: **return** $(\theta, \mathbf{w})$

---

## E.1 COMPLEXITY OF OUR PROPOSED METHOD

The complexity of each step in Algorithm 1 for each model can be written as:

- Extract features: $\mathcal{O}(|\theta|)$

- Fast nearest neighbour search $\approx \mathcal{O}(K \ln M)$ (Johnson et al., 2019) or just $\mathcal{O}(\ln M)$ with GPU

Table 6: The notations used in the complexity analysis.

| Notations | Description |
|---|---|
| $|\theta|$ | the number of model's parameters |
| $M$ | the total number of training samples |
| $C$ | the number of classes |
| $K$ | the number of nearest neighbours |
| $N$ | number of noisy labels per training samples (e.g., $N = 2C - 1$) |
| $L$ | the set of $N$ noisy labels per sample (applicable to ours only) |
| $n_{\text{osqp}}, n_{\text{em}}, n_{\text{em}}$ | the number of iterations used in optimisation |

- Finding similarity matrix $\mathbf{A}$ in (4) with OSQP (Blondel et al., 2022): $\approx \mathcal{O}(n_{\text{osqp}} K d)$, where: $n_{\text{osqp}}$ is the number of iterations and $d$ is the dimension of $X$

- Sampling $L$ sets of $N$-categorical samples where $N \geq 2C - 1$ (in parallel for $N$): $\mathcal{O}(LC^2)$

- Running EM: $\mathcal{O}(n_{\text{em}} N C) \approx \mathcal{O}(n_{\text{em}} C^2)$.

Thus, the complexity of Algorithm 1 per iteration is: $\mathcal{O}(2|\theta| + 2\ln M + 2n_{\text{osqp}} K d + 2(L + n_{\text{em}})C^2)$ since $C \ll M, d$. Nevertheless, the most expensive operation is the sampling that generates additional noisy labels to perform EM with a quadratic complexity in terms of the number of classes $C$. To facilitate the comparison with existing works, we provide a summary of their complexity in Table 2. In general, our method has a higher complexity compared to DivideMix and HOC due to its nature of re-labelling data. The bottle-neck of our proposed method lies at the sampling where $L$ sets of $N$-trial multinomial noisy labels are generated.

### E.2 COMPLEXITY OF DIVIDEMIX

The complexity of DivideMix for each model can be presented in

Table 7: Running time complexity of data processing in DivideMix

| Step | Complexity | Comment |
|---|---|---|
| Cluster with Gaussian mixture model | $\mathcal{O}(M)$ | |
| Augment data | $\mathcal{O}(n_{\text{augment}} \frac{M}{B} d)$ | vectorise over each mini-batch |
| Average prediction loss | $\mathcal{O}(|\theta| + C + M)$ | parallel forward pass |
| Refine ground truth | $\mathcal{O}(\frac{M}{B} C)$ | vectorise over each mini-batch |
| Co-guessing | $\mathcal{O}(2|\theta| + 2M)$ | |
| Sharpen guessed labels | $\mathcal{O}(\frac{M}{B} C)$ | vectorise over each mini-batch |
| **Total per model** | $\mathcal{O}(3|\theta| + (2 + \frac{1}{B}(n_{\text{augment}} d + 2C))M + C)$ | |

Because the number of class $C$ is small compared to the number of samples $M$ or the sample dimension $d$, one can simplify the complexity of the pre-processing step in dual-model DivideMix as follows:

$$\mathcal{O}\left(6|\theta| + \left[4 + \frac{2}{B}(n_{\text{augment}} d + 2C)\right] M\right). \tag{24}$$

### E.3 COMPLEXITY OF HOC

The running time complexity of HOC (Zhu et al., 2021b) is presented in Table 8.

Table 8: Running time complexity of HOC

| Step | Complexity | Comment |
|---|---|---|
| Extract representation | $\mathcal{O}(|\theta|)$ | assume $d, M \ll |\theta|$ |
| Get 2-NN | $\mathcal{O}(2\ln M)$ | |
| Count frequency | $\mathcal{O}(3M)$ | |
| Solve for transition matrix | $\mathcal{O}(n_{\text{iter}} C^2)$ | |
| **Total** | $\mathcal{O}(|\theta| + 3M + 2\ln M + n_{\text{iter}} C^2)$ | |

### E.4 ACTUAL RUNNING TIME

We also provide the running time in practice for these methods in Table 9. Our proposed method takes longer time to run than DivideMix or HOC. For DivideMix, it relies on the small-loss hypothesis to separate which samples are clean or noisy. The bottle-neck in DivideMix properly lies at the dual models used to avoid confirmation bias. For HOC, it relies on nearest-neighbours to obtain

higher-order statistics to determine the transition matrix of interest. That explains why it is the most efficient method. However, the trade-off is that it relies on the class-dependent instant-independent assumption to determine a single transition matrix. That might deteriorate the performance when such an assumption does not hold. For our method, it has the two bottle-necks of DivideMix (2 models) and HOC (nearest-neighbour search). In addition, it also requires to sample a large number of categorical samples. That explains why the method has a longer running time compared to DivideMix and HOC. In practice, we use 2 GPUs and hence, reduce the running time to 50 percent. The reported results in Table 9 are multiplied by 2 (i.e., GPU-h) to be fair when comparing with DivideMix and HOC.

Table 9: Running time of some LNL methods.

| Method | Time (GPU-h) |
|---|---|
| DivideMix - CIFAR-10 | 6.45 |
| HOC - CIFAR-10 | 2.65 |
| Ours - CIFAR-10 | 6.14 |
| Ours - CIFAR-100 | 19.17 |

## F    REDUCING NUMBER OF NOISY LABELS

As mentioned in Section 3.2.2, the space of a noisy label $\hat{Y}$ of an instance $X$ in practice is not often arbitrary (e.g., does not necessary in $\{1, \ldots, C\}$), but may be in a much small set with $C_0$ classes, where $C_0 \ll C$. In that case, the noisy label distribution is no longer a dense mixture of $C$ multinomial distributions, but it is sparse with only $C_0$ components. By exploiting this observation, we can reduce the complexity of the proposed method. In particular, the $C$-component multinomial noisy label distribution, $p(\hat{Y}|X, Y)$, obtained through the EM algorithm is truncated to a $C_0$-component mixture where $C_0 \ll C$. The approximation can be summarised as:

- At the initialisation stage (steps 9 and 10 in Algorithm 1 in Appendix D), the noisy label distribution of each sample is instantiated as a multinomial mixture model of $C_0$ components ($\pi_i$ is $C$-dimensional vector with only $C_0$ non-zero components), where each component still has $C$ categories ($\rho_{ic} \in \Delta_{C-1}$).

- Because the noisy label distribution of each sample is a multinomial mixture model of $C_0$ components, the approximation of noisy label distribution obtained in Eq. (3) results in a multinomial mixture model of $(K + 1)C_0$ components. Such a mixture model can efficiently generate noisy labels with a complexity of $\mathcal{O}((K+1)C_0C)$ given the reasonable size $C_0$.

- The generated noisy labels are passed to the EM algorithm to infer $\pi_i$ and $\rho_i$, which represent the multinomial mixture model $p(\hat{Y}|X = \mathbf{x}_i)$ as shown in Eq. (2). The mixture coefficient $\pi_i$ (also known as the clean label probability) is a $C$-dimensional vector, which may or may not be sparse. We then enforce its sparsity by picking the top $C_0$ components, normalising them to 1, while setting the remaining components to zero. As a result, $\pi_i$ is a $C$-dimensional probability vector with $C_0$ non-negative components. In other words, $p(\hat{Y}|X = \mathbf{x}_i)$ is a multinomial mixture model of $C_0$ components.

For example, in CIFAR-100, although there is a total of 100 classes, these classes are grouped into 20 superclasses, which is equivalent to $C_0 = 5$. This significantly reduces the running complexity of the proposed method by a factor of 20. For web-scale datasets, such as ImageNet, some previous studies partitioned the 1,000 classes into 11 superclasses (Tsipras et al., 2020), reducing 11 times the complexity of the proposed method if it is used for training.

## G    DATASETS AND EXPERIMENT SETTINGS

**Datasets**    For the synthetic instance-dependent label noise setting, we use CIFAR-10 and CIFAR-100 datasets and follow (Xia et al., 2020) to generate synthetic instance-dependent noisy labels. For

the real-world label noise setting, we use three common benchmarks, namely: Controlled Noisy Web Labels (CNWL) (Jiang et al., 2020), mini-WebVision (Li et al., 2017) with additional evaluation on the validation of ImageNet ILSVRC 2012 (Russakovsky et al., 2015), and Animal-10N (Song et al., 2019). For CNWL, we use the web label noise (or red noise) setting where the labels of internet-queried images are annotated manually. For mini-WebVision, we follow previous works that take a subset containing the first 50 classes in the WebVision 1.0 dataset for training and evaluate on the clean validation set. The model trained on mini-WebVision is also evaluated on the clean validation set of ImageNet ILSVRC 2012. Finally, we evaluate the proposed method on Animal-10N dataset that contains 5 pairs of similar-looking animals.

**Models** We follow the same setting in previous studies (Li et al., 2020; Xu et al., 2021) that use PreAct Resnet-18 as the backbone to evaluate the proposed method on CIFAR-10, CIFAR-100 and Red CNWL datasets. For CNWL, input images are resized from 84-by-84 pixel$^2$ to 32-by-32 pixel$^2$ to be consistent with previous evaluations (Xu et al., 2021). For mini-WebVision, we follow the setting in DivideMix (Li et al., 2020) by resizing images to 224-by-224 pixel$^2$ before passing the images into a Resnet-50. For Animal-10N, we follow experiment setting specified in (Song et al., 2019) by training a VGG-19 backbone on 64-by-64 images to obtain a fair comparison with existing baselines.

**Hyper-parameters** The model of interest is warmed-up for 10 epochs with a mini-batch size of 128 training samples and trained for 150 epochs in total. The optimiser used is the stochastic gradient descent (SGD) with a momentum of 0.9 and an initial learning rate of 0.02. The learning rate is decayed following a cosine annealing with a cycle of $10^6$ iterations (gradient update steps). For the priors defined in (5), we assume both priors on the mixture coefficient (or clean label posterior) $\pi_i$ and the probability vector $\rho_{ic}$ of the multinomial components as symmetric Dirichlet distributions with $\alpha = 1.1$ and $\beta = 1.1$. For the nearest neighbours, we first randomly sample a subset of 15,000 samples then perform nearest neighbour search and select the 10 nearest samples for LLC. The parameters $\mu = 0.5$ and $\gamma = 0.95$ are used across all of the experiments.

Our implementation is in JAX (Bradbury et al., 2018) and can be accessed at https://anonymous.4open.science/r/identifiable_label_noise/. All experiments are performed on a computer with a Intel 10th-gen i7 CPU, 32 GB RAM and NVIDIA A6000 GPU.

## H ADDITIONAL RESULTS ON COMMON LNL BENCHMARKS

We provide additional results on the common LNL benchmarks with synthetic instant-dependent label noise on the two datasets CIFAR-10 and CIFAR-100 in Table 10.

## I ADDITIONAL RESULTS OF LABEL CONSISTENCY ON CLASS-DEPENDENT SETTINGS

Table 10: Comparison of prediction accuracy (%) on various instance-dependent label noise rates for CIFAR-10 and CIFAR-100 with different network architectures including self-supervised on the corresponding un-labelled datasets. The majority of results are adopted from (Yao et al., 2021) with † denoting results from their respective papers and * denoting results reported in (Zhu et al., 2021b); the bold numbers denote the maximum mean values across all methods considered. Best result in **bold**, 2nd best in *italics*.

| | CIFAR-10 | | | | CIFAR-100 | | | |
|---|---|---|---|---|---|---|---|---|
| Noise rate | **0.2** | **0.3** | **0.4** | **0.5** | **0.2** | **0.3** | **0.4** | **0.5** |
| Cross-entropy (Yao et al., 2021) | 75.81 | 69.15 | 62.45 | 39.42 | 30.42 | 24.15 | 21.45 | 14.42 |
| mixup (Zhang et al., 2018) | 73.17 | 70.02 | 61.56 | 48.95 | 32.92 | 29.76 | 25.92 | 21.31 |
| Forward (Patrini et al., 2017) | 74.64 | 69.75 | 60.21 | 46.27 | 36.38 | 33.17 | 26.75 | 19.27 |
| T-Revision (Xia et al., 2019) | 76.15 | 70.36 | 64.09 | 49.02 | 37.24 | 36.54 | 27.23 | 22.54 |
| Reweight (Liu & Tao, 2015) | 76.23 | 70.12 | 62.58 | 45.46 | 36.73 | 31.91 | 28.39 | 20.23 |
| Decoupling (Malach & Shalev-Shwartz, 2017) | 78.71 | 75.17 | 61.73 | 50.43 | 36.53 | 30.93 | 27.85 | 19.59 |
| Co-teaching (Han et al., 2018b) | 80.96 | 78.56 | 73.41 | 45.92 | 37.96 | 33.43 | 28.04 | 23.97 |
| MentorNet (Jiang et al., 2018) | 81.03 | 77.22 | 71.83 | 47.89 | 38.91 | 34.23 | 31.89 | 24.15 |
| CausalNL (Yao et al., 2021) | 81.79 | 80.75 | 77.98 | **78.63** | 41.47 | 40.98 | 34.02 | 32.13 |
| CAL (Zhu et al., 2021a)† | 92.01 | - | 84.96 | - | *69.11* | - | 63.17 | - |
| PTD-R-V (Xia et al., 2020)† | 76.58 | 72.77 | 59.50 | 56.32 | 65.33 | 64.56 | 59.73 | 56.80 |
| Peer loss (Liu & Guo, 2020)* | 89.52 ± 0.22 | - | 83.44 ± 0.30 | - | 61.13 ± 0.48 | - | 48.01 ± 0.12 | - |
| $L_{DMI}$ (Xu et al., 2019)* | 88.67 ± 0.70 | - | 83.65 ± 1.13 | - | 57.36 ± 0.97 | - | 43.06 ± 2.39 | - |
| $L_q$ (Zhang & Sabuncu, 2018)* | 85.66 ± 1.09 | - | 75.24 ± 1.07 | - | 56.92 ± 0.24 | - | 40.17 ± 1.52 | - |
| Co-teaching+ (Yu et al., 2019)* | 89.82 ± 0.39 | - | 73.44 ± 0.38 | - | 41.62 ± 1.05 | - | 24.74 ± 0.85 | - |
| JocoR (Wei et al., 2020)* | 88.82 ± 0.20 | - | 71.13 ± 1.94 | - | 44.55 ± 0.62 | - | 23.92 ± 0.32 | - |
| HOC global (Zhu et al., 2021b)† | 89.71 ± 0.51 | - | 84.62 ± 1.02 | - | 68.82 ± 0.26 | - | 62.29 ± 1.11 | - |
| HOC local (Zhu et al., 2021b)† | 90.03 ± 0.15 | - | 85.49 ± 0.80 | - | 67.47 ± 0.85 | - | 61.20 ± 1.04 | - |
| kMEIDTM (Cheng et al., 2022)† | *92.26* | **90.73** | *85.94* | 73.77 | 69.16 | 66.76 | *63.46* | **59.18** |
| IDNT (Wang et al., 2025) | 83.68 ± 0.72 | 79.93 ± 0.65 | 75.57 ± 0.57 | 67.23 ± 0.46 | 54.68 ± 1.38 | 46.93 ± 0.85 | 43.57 ± 0.37 | 38.23 ± 0.76 |
| STMN (Zhang et al., 2024) | 80.10 ± 0.45 | 76.66 ± 2.19 | 71.88 ± 2.19 | 57.14 ± 3.38 | – | 42.65 ± 0.49 | – | 31.12 ± 0.63 |
| **Ours** | **92.39 ± 0.85** | *90.14 ± 1.22* | 85.78 ± 1.27 | 62.07 ± 1.64 | 69.02 ± 1.44 | *66.80 ± 1.92* | 60.85 ± 1.95 | 41.42 ± 1.30 |
| **Ours (DINO)** | 91.16 ± 0.64 | 89.67 ± 1.13 | **86.85 ± 1.80** | *76.03 ± 3.68* | **75.45 ± 0.94** | **73.69 ± 1.23** | **70.32 ± 1.62** | *58.02 ± 2.01* |

Table 11: Label consistency for class-conditioned (or asymmetric noise) label noise on CIFAR-10.

| Method | Noise rate | | |
|---|---|---|---|
| | **0.2** | **0.3** | **0.4** |
| Cross-entropy | 82.35 | 78.14 | 72.02 |
| F-correction | – | – | 83.10 |
| Ours | 90.21 | 87.82 | 79.14 |
| Ours (with SimCLR) | 92.60 | 89.07 | 84.59 |

