# OpenReview forum: "Identifiability in Noisy Label Learning: A Multinomial Mixture Modelling Approach"
_ICLR.cc/2026/Conference — Submitted to ICLR 2026_

### Official Review · Reviewer_apZK · 2025-10-31

**Soundness:** 2
**Presentation:** 2
**Contribution:** 2
**Rating:** 4
**Confidence:** 3

**Summary:**

This paper introduces a multinomial mixture modelling approach to address the identifiability problem in learning from noisy labels (LNL). The authors theoretically prove that LNL becomes identifiable when each sample has at least 2C−1 independent noisy labels, enabling the unique recovery of clean label distributions without relying on heuristic assumptions. To make this feasible in practice, they propose generating additional pseudo noisy labels from nearest neighbours and applying an Expectation–Maximization algorithm to infer clean labels. Extensive experiments on synthetic, web-controlled, and real-world noisy datasets demonstrate that the proposed method accurately estimates clean labels and achieves performance competitive with state-of-the-art LNL techniques.

**Strengths:**

1. The proposed theorem on the identifiability condition is interesting and theoretically sound.
2. The experimental results demonstrate the effectiveness of the proposed approach in improving model robustness under noisy settings.

**Weaknesses:**

1. To be honest, I do not catch up the meaning of the task setting. The paper assumes that each sample is associated with multiple noisy labels, but in practical training datasets, it is rare for a single sample to have multiple annotations. This seems inconsistent with the practicality the authors aim to address.
2. Requiring 2C−1 additional noisy labels per sample would result in a prohibitively large workload.
3. It is not clear whether there are additional constraints for the identifiability condition—such as the i.i.d. assumption for the noisy samples. For instance, what happens if the 2C−1 noisy labels all belong to the same category?
4. The experimental setup lacks sufficient detail. I am a bit confused about what the accuracy in Tables 3 and 4 represents—does it refer to the accuracy of clean label identification, or to the image classification accuracy of the model trained on the selected clean samples? If it is the former, what about the recall metric? If it is the latter, what is the clean identification accuracy?
5. In Figure 1, only classification accuracy is reported. Since classification accuracy can be influenced by various factors (e.g., data scaling effects), directly showing the curve of clean-label identification accuracy would more clearly demonstrate the proposed identifiability condition.

**Questions:**

Please refer to weaknesses.

---

> ### Author Response · Authors · 2025-11-20
> **Theoretical condition requires multiple noisy labels, while there is only one noisy label in practice**
>
> Note that many large-scale datasets initially collect multiple annotations per sample, often via crowdsourcing platforms. However, these annotations are typically aggregated into a single consensus label before public release, and there is no guarantee that this consensus label accurately reflects the true class, especially under high noise or subjective labelling conditions. Our theory provides a principled answer to the question of how many labels are needed for identifiability: at least $2C-1$ i.i.d. noisy labels per instance, which current aggregation practices do not address.
>
> In practice, our approach leverages the single available label per sample in the datasets to approximate the underlying noisy-label distribution, enabling the generation of additional i.i.d. noisy labels without requiring extra manual annotations. This makes the method applicable even when only one label per instance is provided, aligning theoretical insights with practical constraints.

---

> ### Author Response · Authors · 2025-11-20
> **Requesting at least 2C - 1 noisy labels per sample leads to large workload**
>
> We fully agree that this theoretical requirement is impractical in real-world settings. This is precisely why Section 3.2 introduces a practical method that generates additional noisy labels automatically by estimating $\Pr(\hat{Y}|X)$ and sampling from it, without requiring extra manual annotations. In practice, our method requires only one noisy label per training sample.

---

> ### Author Response · Authors · 2025-11-20
> **Constraints for the identifiability condition**
>
> The identifiability condition we prove in Section 3.1 requires only that the noisy labels are i.i.d. without any additional constraints. If all $2C - 1$ labels fall into the same category as mentioned by the reviewer, then the estimated noisy-label distribution $\Pr(\hat{Y}|X)$ is highly concentrated (close to 1 for that class). In such cases, the instance effectively behaves like an "anchor point" (Liu \& Tao, 2015), which strongly indicates its true class. This scenario does not violate our identifiability condition. It simply reflects an extreme case where the distribution provides little uncertainty.

---

> ### Author Response · Authors · 2025-11-20
> **Details of the experiments**
>
> Due to space constraints, detailed experimental setup is provided in Appendix G. The accuracies in Tables 3 and 4 refer to classification accuracy on the *predefined* clean test sets (similar to the standard supervised learning with clean labels), not on selected samples.
>
> For clean-label identification during training, Fig. 1 (middle) reports the accuracy of pseudo-clean labels inferred by our method compared to ground truth labels on the training set. Note that these clean labels are never observed by the models during training.

---

> ### Author Response · Authors · 2025-11-20
> **Demonstrate curve of clean label identification**
>
> Fig. 1 (middle) already includes the clean-label identification curve during training. In particular, the orange curve represents the percentage of correctly identified clean labels compared to ground truth of training samples. This metric directly demonstrates the effectiveness of our identifiability-based approach. We will make this clearer in the caption and text to avoid confusion.

---

### Official Review · Reviewer_VsSa · 2025-10-31

**Soundness:** 2
**Presentation:** 3
**Contribution:** 2
**Rating:** 4
**Confidence:** 3

**Summary:**

This paper addresses the long-standing issue of identifiability in learning from noisy labels (LNL). The authors show that, under a multinomial mixture modeling approach, the LNL problem becomes identifiable if at least $2C-1$ independent and identically distributed (i.i.d.) noisy labels are available per instance (where $C$ is the number of classes). As manually acquiring such redundancy is impractical, the paper proposes estimating additional noisy labels via nearest-neighbour augmentation in feature space. Then the paper use an Expectation-Maximisation (EM) algorithm to estimate the clean label distributions. This algorithm works on the mixture model. The experiments show strong results on both synthetic and real-world datasets. This paper also ran many ablation studies. These studies back up our theoretical ideas and design decisions.

**Strengths:**

- **Theoretical Contribution:** The paper provides a precise, transparent identifiability condition for the LNL problem via multinomial mixture models, generalizing and clarifying prior results by relating the required number of noisy labels per sample to the number of classes in the task (Section 3.1, Table 1).
- **Methodological Novelty:** This paper proposes a practical, systematic way to generate the required number of noisy label replicates using feature-space nearest neighbours—trading off annotation quality for quantity, which is substantiated both theoretically and empirically .
- **Empirical Validation:** The paper shows strong results on several datasets, including CIFAR-10N, CNWL, mini-WebVision, and Animal-10N. It also tests the method under different kinds of label noise. These experiments prove the approach is reliable. Even when compared to current top methods, it holds up well.
- **Clarity and Reproducibility:** The paper gives full math details, especially in Appendix B. It clearly explains why each modeling choice was made. It also talks openly about the tradeoffs in the design. All the practical details are included.
- **Figure Engagement:** The ablations in **Figure 1** sharply illustrate the phase transition as the number of noisy labels or neighbours increases, validating both theoretical claims (left subfigure: test accuracy saturates at $N = 2C-1$) and practical effect of relabeling (middle: % clean after EM).

**Weaknesses:**

1. **i.i.d. Sampling Validity and Empirical Support:** Although the paper claims that additional noisy labels from the approximation $\tilde{P}(\hat{Y}|X)$ are i.i.d. by construction (Section 3.2.1), this rests on the functional form of the mixture, and the empirical validation is limited. While Table 5 indicates a high degree of label consistency among neighbours on CIFAR-10N, it does not quantify the residual dependencies or their downstream effect on identifiability for more structured or heavily corrupted labels.
2. **Computational Complexity for Large C:** The method’s sampling and EM steps incur $\mathcal{O}(C^2(K+1))$ complexity per sample (Table 2, Appx E), which becomes prohibitive as $C$ grows (e.g., ImageNet-scale tasks). The practical efficacy of the C0-sparsification trick (Appendix F) is not deeply analyzed or benchmarked. The bottlenecks, highlighted in the right subfigure of **Figure 1**, may limit scalability and have not been fully benchmarked against distributed or alternative data-driven LNL schemes.
4. **Potential Overfitting to Benchmark Settings:** The method depends heavily on pretrained feature extractors and heavy preprocessing—such as SimCLR or DINO—in many experiments. This might unintentionally leak information or lead to biased results. The paper tries to reduce this risk by using co-teaching (Section 3.2.2). However, it’s not clear how much the good performance comes from strong self-supervised features. It’s also uncertain whether the approach would work as well in areas where those kinds of features aren’t as helpful.
5. **Algorithmic Details and Practical Usability:** The nearest-neighbour scheme, use of LLC, and the stochasticity of the sampling routine, while justified, would benefit from more explicit sensitivity analyses and implementation guidance. Algorithm 1, as presented, leaves choices of $K$, $L$, and ensemble size to heuristics; there is limited discussion on tuning strategies for practitioners.

**Questions:**

1. **On i.i.d. Label Assumption:** Can the authors provide more rigorous empirical or theoretical support for the i.i.d. nature of generated noisy labels via nearest neighbours—especially in high-dimensional, structured data? Specifically, have they measured mutual information or entropy between neighbour-based samples, and how does this affect identifiability guarantees?
2. **Scalability:** How does the proposed method scale as $C$ increases? Could the authors provide more benchmarks (or at least detailed running time/memory analysis) for substantially larger $C$ (e.g., full ImageNet), and is the C0-sparsification sufficient for realistic streaming or web-scale settings?
3. **Non-Uniformity/Imbalance:** How does the method handle class imbalance or structured non-uniformity in annotator error rates? Is the mixture modeling robust under such misspecification? Are there modifications/extensions that could strengthen performance in such cases?
4. **Sensitivity to Hyperparameters:** Could the authors detail or provide ablations on the influence of $K$ (number of neighbours) and $L$ (number of pseudo-label samples) in more challenging benchmarks? Are there principled ways to set or automatically tune these hyperparameters (see Section 4.3 and Algorithm 1)?
5. **Algorithmic Reproducibility:** Is there publicly available implementation for the entire EM and sampling routine (including feature extraction and pre-processing), and are there resources to support scaling to multi-GPU/distributed environments?
6. **Extensions to Partial Labeling and Non-i.i.d. Annotators:** The Missing Related Works section includes several approaches to partial/noisy labelers. Could the proposed approach adapt to scenarios with partial, subjective, or correlated annotators?

---

> ### Author Response · Authors · 2025-11-20
> **i.i.d. assumption and measuring mutual information between neighbours to quantify how it affects indentifiability guarantee**
>
> We re-iterate that the identifiability condition claimed in Section 3.1 only requires that at least 2C - 1 noisy labels are i.i.d., meaning that:
>  - they are from the same underlying distribution $\Pr(\hat{Y} | X = \mathbf{x})$ of an instance $\mathbf{x}$ (i.e., identically distributed), and
>  - knowing one noisy label will not influence or correlate to the knowledge of knowing another noisy label (i.e., independently distributed).
>
> The condition does not require any independence or correlation between neighbouring samples. Neighbouring samples are used solely in our practical method in Section 3.2 to estimate the noisy label distribution $\Pr(\hat{Y} | X = \mathbf{x})$ for a given input $\mathbf{x}$. After this estimation, we generate at least $2C - 1$ noisy labels by sampling i.i.d. from this estimated distribution. Therefore, the independently and identically distributed nature of the generated labels is guaranteed by the sampling procedure itself. The neighbours influence only the estimation of the distribution $\Pr(\hat{Y} | X)$, not the identically and independently distributed nature of the samples drawn from it. We will make this clarification more explicit in line 240 of our revision.
>
> **Mutual information between neighbouring samples** Mutual information is defined between two random variables to quantify their dependence or correlation. In our setting, the training samples (including neighbours) are realisations of the same underlying random variable $X$ (or ($X, \hat{Y}$)) drawn from the data-generating distribution. They are not distinct random variables with separate distributions. Consequently, computing mutual information between neighbours is not meaningful in this context. It does not assess the independence of the sampled labels because the i.i.d. property pertains to the sampling from $\Pr(\hat{Y} | X)$, which is i.i.d. by construction.

---

> ### Author Response · Authors · 2025-11-20
> **Scalability with the number of classes C**
>
> We appreciate the concern about scalability. It is important to emphasise that the primary goal of the paper is to theoretically derive the identifiability condition for noisy-label learning, showing that at least $2C-1$ i.i.d. noisy labels per instance are required for identifiability (Sec. 3.1). This result is fundamental and independent of implementation details because it clarifies when clean labels can be recovered without heuristic priors.
>
> That said, we also provide a practical algorithm (Sec. 3.2) and analyse its scalability. Appendix E details: *complexity* and *runtime measurements* (in GPU-hour), confirming that the main bottleneck is quadratic in $C$. To address this, Appendix F introduces $C_{0}$-sparsification, reducing complexity from $\mathcal{O}(C^2)$ to $\mathcal{O}(C_{0}C)$ by leveraging the empirically supported observation that noisy labels typically occur within a small candidate set (Han et al., 2018a). This assumption holds in large-scale web data and enables practical deployment, which is evidenced in our paper:
>  - Experiments already include the largest noisy-label benchmark in terms of classes, such as CIFAR-100 with 100 classes, where the method remains practical (Table 9).
>  - For larger $C$, $C_{0}$-sparsification (Han et al., 2018a) combined with approximate nearest-neighbour search (TPU-KNN) (Chern et al., 2022) scales efficiently.
>
> We will make these results more prominent and add projected complexity for web-scale scenarios in the revision.

---

> ### Author Response · Authors · 2025-11-20
> **Non-uniform and imbalanced data**
>
> Our work focuses on the theoretical identifiability condition for noisy label learning (Section 3.1). To isolate this factor and avoid confounding effects, we use balanced-class settings in all experiments. This choice ensures that observed performance differences stem from identifiability rather than imbalance. The imbalanced setting will be considered in our future research.

---

> ### Author Response · Authors · 2025-11-20
> **Sensitivity analysis for hyper-parameters K (number of nearest neighbours) and L (number of pseudo-labels generated)**
>
> We have provided the analysis on the number of nearest neighbours and the number of pseudo-labels generated in Fig. 1. In particular:
>  - Fig. 1 (left): effect of varying the number of pseudo-labels per sample $L$, and
>  - Fig. 1 (right): effect of varying different numbers of nearest neighbours $K$.

---

> ### Author Response · Authors · 2025-11-20
> **Algorithmic reproducibility**
>
> We provide our implementation at the link in line 1102 of our paper. The anonymous repository in that link includes the complete pipeline for feature extraction, nearest-neighbour search, sampling, and EM-based inference.
>
> The EM algorithm itself is general and widely used. Our implementation is tailored to the multinomial mixture setting described in Appendix B.
>
> For scalability, we already employ parallel training on two GPUs, as reported in Appendix E.4. Extending to multi-GPU or distributed environments is straightforward using standard frameworks, such as Jax or PyTorch.

---

> ### Author Response · Authors · 2025-11-20
> **Extensions to Partial Labeling and Non-i.i.d. Annotators**
>
> Our primary objective is to establish a formal identifiability condition for standard noisy-label learning (Sec. 3.1), proving that at least $2C-1$ i.i.d. noisy labels per instance are necessary for identifiability. This result is fundamental and deliberately assumes no annotator-specific structure beyond independence, ensuring clarity and generality. Both the identifiability result and the proposed practical method are designed for the conventional noisy-label learning setting and do not directly extend to partial-label learning or scenarios with non-i.i.d. annotators. These settings introduce additional latent dependencies and correlation structures that require different modelling assumptions and identifiability analyses, which fall outside the scope of this work. We will acknowledge these connections in the related work section and highlight them as promising directions for future research.

---

### Official Review · Reviewer_vNTX · 2025-10-31

**Soundness:** 3
**Presentation:** 3
**Contribution:** 3
**Rating:** 6
**Confidence:** 3

**Summary:**

The paper tackles the fundamental issue of identifiability in learning with noisy labels (LNL).
The authors demonstrate that when each sample is annotated with at least 2C−1 i.i.d. noisy labels (where C is the number of classes), the true clean-label distribution becomes identifiable under a multinomial mixture model.
Since collecting that many labels per sample is infeasible in practice, the authors propose a practical algorithm that approximates i.i.d. noisy labels using KNN and LLC, followed by an EM procedure to recover clean posterior estimates.
Extensive experiments on multiple benchmarks show that this surrogate approach is both theoretically justified and empirically effective.

**Strengths:**

1. Novel and intuitive idea.

The identifiability-based perspective is conceptually novel in the LNL literature. The proposed KNN-based generation of “pseudo i.i.d. noisy labels” is both intuitive and innovative, providing a clear interpretation and offering potential for many future extensions.

2. Solid empirical results.

The method achieves competitive and stable performance across real-world noisy datasets. Despite its theoretical nature, the approach remains practical and delivers comparable results to state-of-the-art robust learning methods.

**Weaknesses:**

1. Practical labeling feasibility.

In practice, collecting 2C−1 i.i.d. labels per instance is unrealistic, so the framework relies entirely on pseudo-label generation. My main concern is the algorithm’s dependency on the quality of synthetic labels: when the neighborhood structure is unreliable or features are biased, the identifiability assumption may not hold.

(Optional, minor) The approach might also be computationally demanding for large C, as both the KNN and EM steps scale quadratically with the number of classes.

**Questions:**

1. In extremely noisy scenarios, generating high-quality pseudo i.i.d. labels via KNN may be difficult.
Did the authors analyze the effect of noise level on KNN reliability?
For example, is there an empirical study (similar to Table 5) showing how varying noise levels impact the accuracy or consistency of KNN-based pseudo-label construction (I mean various noise levels and various ks)?

2. Have the authors considered using feature denoising or representation refinement (e.g., pre-trained or self-supervised embeddings) to mitigate the dependence on feature quality during pseudo-label generation?

---

> ### Author Response · Authors · 2025-11-20
> **Impact of higher noise rates to the quality of neighbour consistency**
>
> We agree that noise level influences the quality of nearest-neighbour search, especially during the warm-up phase. Here, we extend the empirical results in Table 5 on label consistency (agreement between a sample’s clean label and those of its neighbours) by adding Cifar-10N "worst label" setting (corresponding to 40 percent noise rate) as follows"
>
> | | K = 10 | | K = 100 | |
> |---|---|---|---|---|
> |Settings | Training directly until converged | Warmup | Training directly until converged | Warmup |
> | 3 labels (2\% noise) | 97.96 | 65.49 | 62.10 | 55.25 |
> | Random 1 (17\% noise) | 89.04 | 60.95 | 54.29 | 42.52 |
> | Random 2 (18 \% noise) | 88.68 | 59.70 | 54.36 | 42.58 |
> | Random 3 (18\% noise) | 88.79 | 59.73 | 54.79 | 42.48 |
> | Worst label (40\% noise) | 53.73 | 45.05 | 43.15 | 33.65 |
>
> The results show that higher noise rates reduce label consistency. In addition, using too many neighbours early in training without proper weighting can be detrimental because naively adding more neighbours increases the chance of including mislabelled samples. At high noise rates, the probability that some neighbours are in different classes grows quickly. Furthermore, adding more neighbours will include samples that are less similar in the feature space, resulting in a bias and less label consistency. That is why we do not simply average over the labels of nearest neighbours but weight by a factor that is inversely proportional to their distances as shown in Eq. (3). For other datasets, we mitigate this effect further by leveraging self-supervised representations (e.g., SimCLR) or pre-trained models such as DINO to improve feature quality and neighbour reliability.

---

> ### Author Response · Authors · 2025-11-20
> **Using feature denoising or representation refinement**
>
> We agree that feature quality is critical for nearest-neighbour search. To address this, our evaluation already incorporates representation refinement. For example, we use a SimCLR model trained on the dataset CNWL, and the DINO pre-trained embeddings on mini-WebVision and Animal-10N.

---

### Official Review · Reviewer_84f6 · 2025-11-01

**Soundness:** 3
**Presentation:** 3
**Contribution:** 3
**Rating:** 6
**Confidence:** 3

**Summary:**

This paper studies the foundational identifiability problem in learning from noisy labels (LNL). It establishes that the standard single-label LNL setting is non-identifiable in theory, meaning the clean label distribution cannot be recovered without additional assumptions. The key contribution is proving that if each instance has at least 2C−1 i.i.d. noisy labels (where C is the number of classes), then clean labels are identifiable when modeling noisy labels as a multinomial mixture. Extensive experiments on synthetic and real-world noisy-label benchmarks support the theoretical claims, showing competitive or improved performance relative to state-of-the-art baselines such as DivideMix, HOC, and others.

**Strengths:**

- This paper provides a clear identifiability condition for LNL: at least 2C−1 i.i.d. noisy labels needed.
- This paper offers a clean alternative to full-rank transition matrix assumptions used in prior work.
- Nearest-neighbor-based generation of pseudo noisy labels is elegant and interpretable.
-This paper includes complexity analysis and ablation experiments clarifying where the cost arises (sampling and EM).

**Weaknesses:**

- This paper relies on good features for nearest-neighbor graph; co-teaching partially mitigates but tight coupling remains.
- This paper argues generated labels are i.i.d. when sampled from estimated distributions, but correlation in neighbor graph may violate strict independence.
- Behavior in class-dependent noise not explicitly reported.

**Questions:**

N/A

---

> ### Author Response · Authors · 2025-11-20
> **Labels are required to be i.i.d. when sampled from estimated distributions, but correlation in neighbor graph may violate strict independenc**
>
> We note that the identifiability condition claimed in Section 3.1 requires the noisy labels of one instance must be i.i.d., meaning that:
>  - they are from the same distribution (i.e., identically distributed), and
>  - the value of one noisy label does not influence or correlate to the value of another noisy label of the same instance (i.e., independently distributed).
>
> The condition does not have any requirements on the estimation of the noisy label distribution $\Pr(\hat{Y} | X)$ or the dependence between neighbouring samples. While the approximation uses neighbour information, this affects only the parameter estimation of $\Pr(\hat{Y} | X)$. Once the distribution is estimated via Eq. (3), the sampling process is i.i.d. This distinction is explicitly noted in Section 3.2.1 - line 240. We will clarify further in our revision by stating that at least 2C - 1 noisy labels must be i.i.d. given the noisy label distribution $\Pr(\hat{Y} | X)$.

---

> ### Author Response · Authors · 2025-11-20
> **Additional results on class-dependent noise**
>
> We note that our theoretical result in Section 3.1 and the proposed method in Section 3.2 hold regardless of noise type. In the submission, we evaluate on real-world datasets (e.g., Cifar-10N, CNWL, mini-WebVision, and Animal10N). Here, we provide additional results on the prediction accuracy of the class-dependent label noise (also known as asymmetric noise) evaluated on Cifar-10 reported using the last iteration checkpoint as follows:
>
> | Method | Noise rate | | |
> |---|---|---|---|
> |  | 0.2 | 0.3 | 0.4 |
> | Cross-entropy | 82.35 | 78.14 | 72.02 |
> | F-correction | _  | _  | 83.10 |
> | Ours | 90.21 | 87.82 | 79.14 |
> | Ours (with SimCLR) | 92.60 | 89.07 | 84.59 |
>
> We will add these results into our revision.

---

### Author Response · Authors · 2025-12-03
**Summary of changes**

We thank the reviewers for their constructive feedback. We have incorporated these suggestions into our submission and uploaded the revision to Open Review with changes being highlighted in blue colour. Below is the summary of changes we have made:
 - We clarify further the Identifiability Condition by specifying that 2C - 1 noisy labels are sampled from the noisy label distribution $\Pr(\hat{Y} | X)$.
 - We clarify the validity of the practical method presented in Section 3.2. In particular, the Identifiability Condition only assumes the i.i.d. noisy labels for each training sample. It does not have any requirements on the estimation of the noisy label distribution $\Pr(\hat{Y} | X)$ or the dependence between neighbouring samples.
 - We added additional results of label agreement on CIFAR-10N in Table 5.
 - We added the projection on the running time complexity of our practical method in Appendix F.
 - We also added label consistency for class-conditioned label noise on CIFAR-10.

---

### Meta-Review · Area_Chair_mpZA · 2026-01-06

**Summary:**

This paper studies the identifiability problem in learning from noisy labels (LNL) and proves a theoretically clean result: under a multinomial mixture model, clean labels are identifiable if each instance has at least 2C-1 is the number of classes. To bridge theory and practice, the authors propose a nearest-neighbour-based procedure to estimate a noisy-label distribution and then sample pseudo i.i.d. noisy labels, followed by EM-based inference. Reviewers generally agreed that the identifiability result is interesting and technically sound, and that the empirical results on standard noisy-label benchmarks are competitive. However, the overall assessment converged on the view that the core theoretical condition is impractical, and the proposed surrogate mechanism introduces strong dependencies on feature quality, modelling assumptions, and computational scalability.

**Reviewer Concerns:**

The rebuttal successfully addressed several clarifications and completeness issues. In particular, the authors clarified the precise meaning of the i.i.d. requirement (i.i.d. given the estimated noisy-label distribution), added experiments under class-dependent (asymmetric) noise, provided additional analyses on neighbour consistency under higher noise rates, clarified experimental metrics, and pointed to implementation details and scalability heuristics (e.g., C0-sparsification, approximate KNN). These responses improved the paper’s clarity and internal consistency.

However, substantive concerns remain unresolved. First, multiple reviewers remained unconvinced that the practical algorithm genuinely satisfies the spirit of the i.i.d. assumption required by the identifiability theorem, as the estimated noisy-label distribution itself depends heavily on correlated neighbour structures and strong feature representations. While the authors argue that i.i.d. sampling is guaranteed once the distribution is estimated, the reliability and bias of that estimate—especially under high noise or weak features—remain insufficiently characterized. Second, the practical relevance of the theoretical result is limited, since the requirement of 2C−1 noisy labels per instance is far from realistic, and the surrogate generation procedure effectively shifts the burden to representation quality and heuristic design choices. Third, reviewers raised concerns about scalability and usability for large numbers of classes, reliance on pretrained or self-supervised features, and limited analysis of failure modes beyond balanced benchmark settings.

**Reviewer Scores:**

Reviewer 84f6 (score: 6): While finding the identifiability condition elegant and the experiments solid, this reviewer explicitly noted concerns about the i.i.d. assumption and feature dependence, and stated they “would not mind if the paper is rejected.” After rebuttal, the score would likely remain at 6, without a clear upward revision.

Reviewer vNTX (score: 6): This reviewer viewed the identifiability perspective as novel but expressed persistent doubts about the feasibility and robustness of pseudo-label generation under high noise and unreliable features. The rebuttal added analyses but did not fundamentally resolve these concerns, so the score would remain at 6.

Reviewer VsSa (score: 4): This reviewer raised deeper concerns about the validity of the i.i.d. assumption, scalability to large C, and dependence on strong pretrained features. Although the rebuttal was thorough, these core limitations remain, and the score would remain at 4.

Reviewer apZK (score: 4): This reviewer questioned the practicality of the task formulation, clarity of the experimental setup, and real-world relevance of requiring multiple noisy labels per instance. While some confusion was clarified, the fundamental mismatch between theory and practice persists, so the score would remain at 4.

---

### Decision · Program_Chairs · 2026-01-26

Reject